

# Identification of fibrillogenic regions in human triosephosphate isomerase

Edson N. Carcamo-Noriega and Gloria Saab-Rincon

Instituto de Biotecnología, Departamento de Ingeniería Celular y Biocatálisis,
Universidad Nacional Autónoma de México, Cuernavaca, Morelos, Mexico

## ABSTRACT

**Background.** Amyloid secondary structure relies on the intermolecular assembly of polypeptide chains through main-chain interaction. According to this, all proteins have the potential to form amyloid structure, nevertheless, in nature only few proteins aggregate into toxic or functional amyloids. Structural characteristics differ greatly among amyloid proteins reported, so it has been difficult to link the fibrillogenic propensity with structural topology. However, there are ubiquitous topologies not represented in the amyloidome that could be considered as amyloid-resistant attributable to structural features, such is the case of TIM barrel topology.

**Methods.** This work was aimed to study the fibrillogenic propensity of human triosephosphate isomerase (HsTPI) as a model of TIM barrels. In order to do so, aggregation of HsTPI was evaluated under native-like and destabilizing conditions. Fibrillogenic regions were identified by bioinformatics approaches, protein fragmentation and peptide aggregation.

**Results.** We identified four fibrillogenic regions in the HsTPI corresponding to the $\beta$3, $\beta$6, $\beta$7 y $\alpha$8 of the TIM barrel. From these, the $\beta$3-strand region (residues 59–66) was highly fibrillogenic. In aggregation assays, HsTPI under native-like conditions led to amorphous assemblies while under partially denaturing conditions (urea 3.2 M) formed more structured aggregates. This slightly structured aggregates exhibited residual cross-$\beta$ structure, as demonstrated by the recognition of the WO1 antibody and ATR-FTIR analysis.

**Discussion.** Despite the fibrillogenic regions present in HsTPI, the enzyme maintained under native-favoring conditions displayed low fibrillogenic propensity. This amyloid-resistance can be attributed to the three-dimensional arrangement of the protein, where $\beta$-strands, susceptible to aggregation, are protected in the core of the molecule. Destabilization of the protein structure may expose inner regions promoting $\beta$-aggregation, as well as the formation of hydrophobic disordered aggregates. Being this last pathway kinetically favored over the thermodynamically more stable fibril aggregation pathway.

Corresponding author
Gloria Saab-Rincon,
gsaab@ibt.unam.mx

## INTRODUCTION

The conversion of native soluble proteins into highly structured insoluble fibrillar assemblies is associated with several degenerative pathologies, such as Alzheimer's disease, Huntington's disease and Parkinson's disease, among others (*Chiti & Dobson, 2006*). Currently, more than 30 different proteins are involved in amyloid processes that differ

in their sequence, topology, size and function but share similar structural features in the fibrils formed. A common unit of the cross-$\beta$ spine structure is conserved in fibrillar assemblies, where $\beta$-strands stack to form $\beta$-sheets that grow perpendicular to the fiber axis (*Jahn et al., 2010*; *Knowles, Vendruscolo & Dobson, 2014*; *Westermark, 2005*). Several fibrillogenesis mechanism models have been proposed based on kinetic, structural and morphological data (*Zerovnik et al., 2011*). These mechanisms differ from each other by: (i) the number of aggregation pathways, (ii) number of steps (conformational and oligomeric states of the protein) and (iii) the cooperativity of amyloid assembly (*Calamai, Chiti & Dobson, 2005*; *Dovidchenko, Leonova & Galzitskaya, 2014*; *Gillam & MacPhee, 2013*; *Gosal et al., 2005*; *Powers & Powers, 2008*; *Wu & Shea, 2011*; *Zou et al., 2014*).

Regardless the mechanism followed, amyloid fibrils formation always relies on the intermolecular interactions of the polypeptide main chain. Therefore, all polypeptide chains have the potential to form $\beta$-aggregates. However, not all proteins aggregate into cross-$\beta$ structures in the physiological environment (*Baldwin et al., 2011*). To observe protein aggregation, the native state must usually be destabilized to expose aggregation-prone regions (*Knowles, Vendruscolo & Dobson, 2014*). It is clear that some sequences are more prone to form cross-$\beta$ structures than others (*Jahn & Radford, 2008*). Some experimental and computational studies have identified the inherent properties of the sequence, including the net charge, length, hydrophobicity and secondary structure propensities, as determinants of aggregation (*Fernandez-Escamilla et al., 2004*; *Hills Jr & Brooks 3rd, 2007*; *Maurer-Stroh et al., 2010*). Based on experimental evidence, several informatics approaches have been developed to identify aggregation-prone regions that could allow us to predict protein aggregation propensity.

These aggregation-prone regions are present in a large number of proteins and, in some cases, play a key role in the function or folding of the protein and, therefore, cannot be eliminated. However, evolution has developed protective mechanisms, such as improving solubility, steric hindrance and conformational restriction, to avoid the exposure of aggregation-prone regions (*Richardson & Richardson, 2002*; *Tzotzos & Doig, 2010*). It is clear that most of the protective mechanisms that evolved, are based on increasing the stability of the native state of the protein, suggesting that topologies with higher stabilities are less susceptible to amyloid aggregation (*Baldwin et al., 2011*). Thus, this trait can be selected during evolution, which may explain the rather limited number of folds observed in nature (*Goldstein, 2008*; *Koehl & Levitt, 2002*).

A scaffold that has been recursively recruited during evolution is the TIM barrel $(\beta/\alpha)_8$ (*Wierenga, 2001*). So far, there are no reports of any TIM barrel forming fibrillar aggregates. Nevertheless, there is some evidence that links triosephosphate isomerase (TPI) with amyloid aggregation. In 1999, Contreras and coworkers *(1999)* found a segment (residues 186–218) from *Escherichia coli* TPI that shares 20% identity with the amyloid $\beta$-peptide. When investigated, this fragment was able to assemble into amyloid-like fibrils with affinity to Congo red. In another work, Guix and coworkers found high levels of a nitrated variant of TPI (nitro-TPI) that was induced by amyloid $\beta$-peptide depositions in the brain tissues of Alzheimer's disease patients *Guix et al. (2009)*. Aggregation of *in vitro* nitrated rabbit TPI formed large $\beta$-aggregates with amyloid-like properties that were

able to induce aggregation of the Tau protein. These data indicate a fibrillogenic potential in TIM barrels, moreover in human triosephosphate isomerase (HsTPI) with interest in the physiopathology of Alzheimer's disease. Therefore, we considered imperative to evaluate the *in vitro* propensity of HsTPI to aggregate into fibril conformation and search for fibrillogenic regions in its sequence.

## MATERIALS AND METHODS

### Materials

All peptides were synthesized by Liquid Phase Peptide Synthesis (LPPS) without N- or C-terminal modifications by GenScript USA Inc. (Fig. 3B). The WO1 antibody was generously donated by Dr. Ronald Wetzel from the Department of Structural Biology of the University of Pittsburgh. Hen egg-white lysozyme (HEWL) and all other reagents were purchased from Sigma-Aldrich Co (St. Louis, MO, USA).

### Expression and purification of HsTPI

The vector pET3a-HsTPI was kindly donated by Dr. Gomez Poyou (IFC-UNAM). This vector encodes the sequence of the wild-type HsTPI with a His-tag at the N-terminus followed by a TEV protease recognition sequence. The plasmid was transformed into the *E. coli* BL21-Gold(DE3) strain. Transformed cells were grown in LB medium supplemented with ampicillin at 37 °C until an absorbance of 0.6 at 600 nm was reached. Then, the expression of HsTPI was induced with IPTG at a final concentration of 0.2 mM. Incubation continued at 20 °C for 6 h. The cells were harvested and suspended in buffer A (20 mM sodium phosphate, 150 mM NaCl, pH 7.4). The cell suspension was sonicated (five times for 30 s) and centrifuged at 15,000 g for 30 min. The supernatant was loaded into a Ni-NTA agarose column. The resin was washed with 10 column volumes of buffer A containing 50 mM imidazole. HsTPI was then eluted with 300 mM imidazole in buffer A. The purified HsTPI was dialyzed against buffer A in order to eliminate the imidazole. The His$_{6X}$ tag was cleaved using recombinant His-tagged TEV protease at a ratio of 1:50 (w/w) protease/HsTPI at 4 °C overnight. To remove the His-tagged TEV protease as well as any undigested HsTPI, the mixture was loaded into a Ni-NTA agarose column and washed with buffer A. The effluent from the column containing the cleaved HsTPI was recovered, precipitated with 75% ammonium sulfate and stored at 4 °C. The protein concentration was determined by measuring the absorbance at 280 nm using an extinction coefficient of $\varepsilon = 33{,}460$ M$^{-1}$ cm$^{-1}$. The integrity of HsTPI was checked using a specific activity assay.

### Aggregation assays

Aggregation assays were performed in 1.5-mL Eppendorf tubes containing 1 mL of 100 μM HsTPI previously dialyzed against buffer A supplemented with 0.2% of sodium azide. Aggregation was also performed under slightly denaturing conditions (3.2 M of urea in the same buffer). The tubes containing the enzyme were incubated in a thermomixer (series 22670000; Eppendorf, Hamburg, Germany) at 37 °C and 600 rpm for one week. HsTPI aggregation was followed through time by a turbidimetric assay

at 405 nm and by Thioflavin T (ThT) fluorescence. To do so, protein samples (10 µL) were added to 140 µL of filtered 10 µM ThT in buffer A and the fluorescence emission intensity at 485 nm was recorded using 96-well black clear bottom plates in a Tecan Safire multimode microplate reader at an excitation of 440 nm. After one week of incubation, the aggregates were recovered by centrifugation at 25,000 g for 1 h. The aggregation kinetics were repeated three times with protein originating from different expression and purification batches.

For peptide aggregation assays, desalted freeze-dried peptide was first dissolved. All but the $\beta$4, $\beta$7, and $\beta$8 peptides were soluble in water. The $\beta$4 and $\beta$7 peptides were dissolved in 10% acetic acid while the $\beta$8 peptide was dissolved in 100 mM ammonium hydroxide. After dissolving, peptide solutions were diluted with phosphate buffer, supplemented with 0.02% of sodium azide, to a final concentration of 50 µM and the pH was adjusted to 7.4. Large particles were removed by micro filtration (0.45-µm pore size). A 1 mL sample of each peptide solution in 1.5-mL Eppendorf tubes was sealed and incubated at 37 °C and 600 rpm for 3 weeks. The final ThT fluorescence intensity and green-birefringence with Congo red were measured.

## Congo red birefringence

Congo red binding analysis was conducted by spectrophotometric assay. First, 10 µL of an aggregate sample was added to 140 µL of filtered 5 µM Congo red in PBS. Congo red alone was used as a reference. The mixtures were incubated at room temperature for 30 min. Absorbance spectra from 400 nm to 700 nm were acquired on a Tecan Safire multimode microplate reader blanked with phosphate buffer. A maximum peak at 540 nm was indicative of red-green birefringence. A relative birefringence value was calculated using the ratio of absorbance at 540:490 nm, $b = (\text{abs}_{540\,\text{nm}}/\text{abs}_{490\,\text{nm}})$ based on previous reports (*Frid, Anisimov & Popovic, 2007*; *Klunk, Jacob & Mason, 1999*).

## Dot-blot assay

To confirm cross-$\beta$ structure in the aggregates, a dot-blot assay against the anti-cross-$\beta$ WO1 antibody was performed. First, a 10 µL sample was placed as a drop on a nitrocellulose membrane and allowed to dry. Non-specific binding sites were blocked with 5% (w/v) bovine serum albumin for 1 h at room temperature. The membrane was incubated for 1 h with the WO1 antibody at a dilution of 1:8,000 in phosphate buffer containing 0.05% (w/v) Tween 20 (T-PBS). The unbound primary antibody was washed three times for 10 min with T-PBS. Then, the membrane was incubated for 1 h at room temperature with the secondary antibody (alkaline phosphate conjugated anti-mouse antibody; A3562; Sigma-Aldrich, St. Louis, MO, USA) using a dilution factor of 1:30,000 in T-PBS. The membrane was then washed 5 times with T-PBS for 10 min and revealed using the BCIP®/NBT-Blue Liquid Substrate System for Membranes for 10 min. The colorimetric reaction was stopped with MilliQ water.

## Transmission electron microscopy (TEM)

The final aggregation products were placed on Formvar-coated 200 mesh copper grids for 1 min. The grids were stained for 1 min with 2% (w/v) uranyl acetate and then washed

once with MilliQ water. The images were recorded on a ZEISS transmission electron microscope model LIBRA 120 operating at 120 kV.

### Infrared spectroscopy

Fourier-transform infrared (FTIR) spectra of HsTPI samples were recorded using a Perkin Elmer-Spectrum Rx1 spectrometer equipped with a zinc selenide (ZnSe) Attenuated total reflection (ATR) accessory. Sample treatment and data recording was carried out as previously described (*Shivu et al., 2013*). A total of 256 accumulations at 1 cm$^{-1}$ of resolution were performed in the range of 1,800–1,500 cm$^{-1}$. Water-vapor spectrum was subtracted from all samples spectrum and then spectral intensities were normalized in the 1,630 cm$^{-1}$ peak using the Spekwin32 software. Furthermore, raw spectra in amide I region (1,700–1,600 cm$^{-1}$) were analyzed by second-derivative with PeakFit 4.12 software using the Savitsky-Golay routine.

### Cross-$\beta$ region consensus prediction

Potential fibrillogenic regions were predicted using HsTPI sequence (UniProt ID P60174-1). A consensus prediction was considered to be at least two sequence hits by any of the four different predictors used: FISH-AMYLOID, FOLD-AMYLOID, PASTA 2.0 and AMYLPRED 2. For all servers, the default parameters were used.

### Acid hydrolysis of HsTPI

The chemical cleavage reaction was carried out in 1.5-mL Eppendorf tubes. Ten mg of freeze dried HsTPI was dissolved in 1 mL of 10 mM HCl, 1 mM DTT, pH 2, and incubated at 65 °C for 8 h. After the incubation period, the reaction was cooled on ice and the hydrolysis pattern was analyzed by tricine SDS-PAGE stained with coomassie dye. The hydrolysis products were incubated at 37 °C and 600 rpm for 7 days. The resulting aggregates were washed five times with water and then disaggregated with 7.4 M guanidinium chloride (Gdm-HCl) by mixing overnight at room temperature.

### Mass spectroscopy analysis

The dissolved aggregates were desalted using a SepPack C18 cartridge and analyzed by nanoliquid chromatography and tandem mass spectrometry (nLC-MS/MS) with collision-induced dissociation (CID) on a LTQ-Orbitrap Velos (Thermo-Fisher Co., San Jose, CA, USA) integrated with EASY-nLC II (Thermo-Fisher Co., San Jose, CA). For reverse chromatography, a 25-cm analytical column (750-$\mu$m inner diameter) packed with C18 resin was used in a continuous flow of 400 nL/min in a 10–90% gradient of acetonitrile in 0.1% formic acid over 120 min. All spectra were acquired in a data-dependent mode at a resolution of 60,000 with an $m/z$ range of 300–1,600. Ions with a charge of +2, +3 and +4 were isolated for fragmentation using a normalized collision energy value of 35 and an activation $Q$ value of 0.25.

## RESULTS

### HsTPI aggregation

The $\beta$-aggregation propensity of HsTPI was evaluated by incubation with stirring for 7 days at 37 °C under native-like conditions (HsTPI$_n$). Additionally the incubation was

also carried out in 3.2 M of urea (HsTPI$_{urea}$), a condition slightly destabilizing but still at the beginning of the unfolding transition (*Mainfroid et al., 1996a*; *Mainfroid et al., 1996b*). The kinetics of aggregation was followed by ThT fluorescence (Fig. 1A) and by visible light dispersion monitored at 405 nm (Fig. 1B). A slight increase in the ThT fluorescence is observed for HsTPI$_n$ after 40 h. Nevertheless this low fluorescence intensity was not indicative of β-aggregation; the more drastic increment in turbidity, as indicated by the dispersion of visible light, suggests that it is more likely an interference of non-specific binding of the ThT to disordered aggregates, as displayed by other amorphous assemblies (*Biancalana & Koide, 2010*; *Nielsen et al., 2012*; *Scarafone et al., 2012*). HsTPI$_{urea}$, on the other hand, displayed a fast but small increase in turbidity (Fig. 1B), while ThT fluorescence showed a more significant increase (Fig. 1A). In this case, a short lag phase (about 5 h) is observed with an elongation phase extended up to 7 days without reaching a plateau. According to this data, amorphous aggregation precedes β-aggregation of HsTPI$_{urea}$, which was not complete after 7 days of incubation. Longer incubation was not possible due the loss of protein by adhesion to the tube and microbial contamination despite that sodium azide was added. The TEM images of the final aggregation products of HsTPI$_n$ showed disordered aggregates with a fragmented appearance (Fig. 1C). It is worth to mention that staining conditions were the same for both samples; however, the HsTPI$_n$ sample showed some spots that seem to be overstained. Image analysis shows a very entropic saturation of this micrograph compared to the one from HsTPI$_{urea}$, suggesting that the very dark spots are indeed reflecting a high concentration of protein aggregated. In the case of HsTPI$_{urea}$ the aggregates displayed some elongated structures co-aggregated with clusters of disordered aggregates that seem to be in an incomplete stage of the fibrillogenic pathway. This observation is in good agreement with the slow rate of β-aggregation as detected by ThT fluorescence.

ATR-FTIR was performed in order to evaluate the secondary structure of the aggregates obtained after a week of incubation. The second-derivative of the IR spectrum of salted-out HsTPI shown two maximal peaks around 1,655 and 1,633 cm$^{-1}$ in the amide I region (Fig. 2A). These bands correspond to α-helix and β-sheet structures, respectively. This second-derivative ATR-FTIR spectrum was consistent with spectra of others $(\beta/\alpha)_8$ barrel protein in H$_2$O (*Baldassarre et al., 2011*; *Dong, Huang & Caughey, 1990*; *Huang & Dong, 2003*; *Kong & Yu, 2007*). After one week of incubation under native-favoring conditions, the secondary structure of HsTPI was virtually unchanged suggesting native-like aggregation. In contrast, HsTPI$_{urea}$ showed an increase of β-structure (1,624 cm$^{-1}$ band) upon aggregation. Some residual non-β secondary structure was maintained around 1,656 cm$^{-1}$ indicating that β-aggregation was not complete. Recent studies have sighted a clear tendency in the formation of new β-structure formation upon aggregation despite the nature of the aggregates (*Shivu et al., 2013*; *Wang et al., 2010*). In this regard, cross-β structure was further confirmed for HsTPI$_{urea}$ aggregates by the recognition of the anti-cross-β WO1 antibody (*O'Nuallain & Wetzel, 2002*) (Fig. 2B).

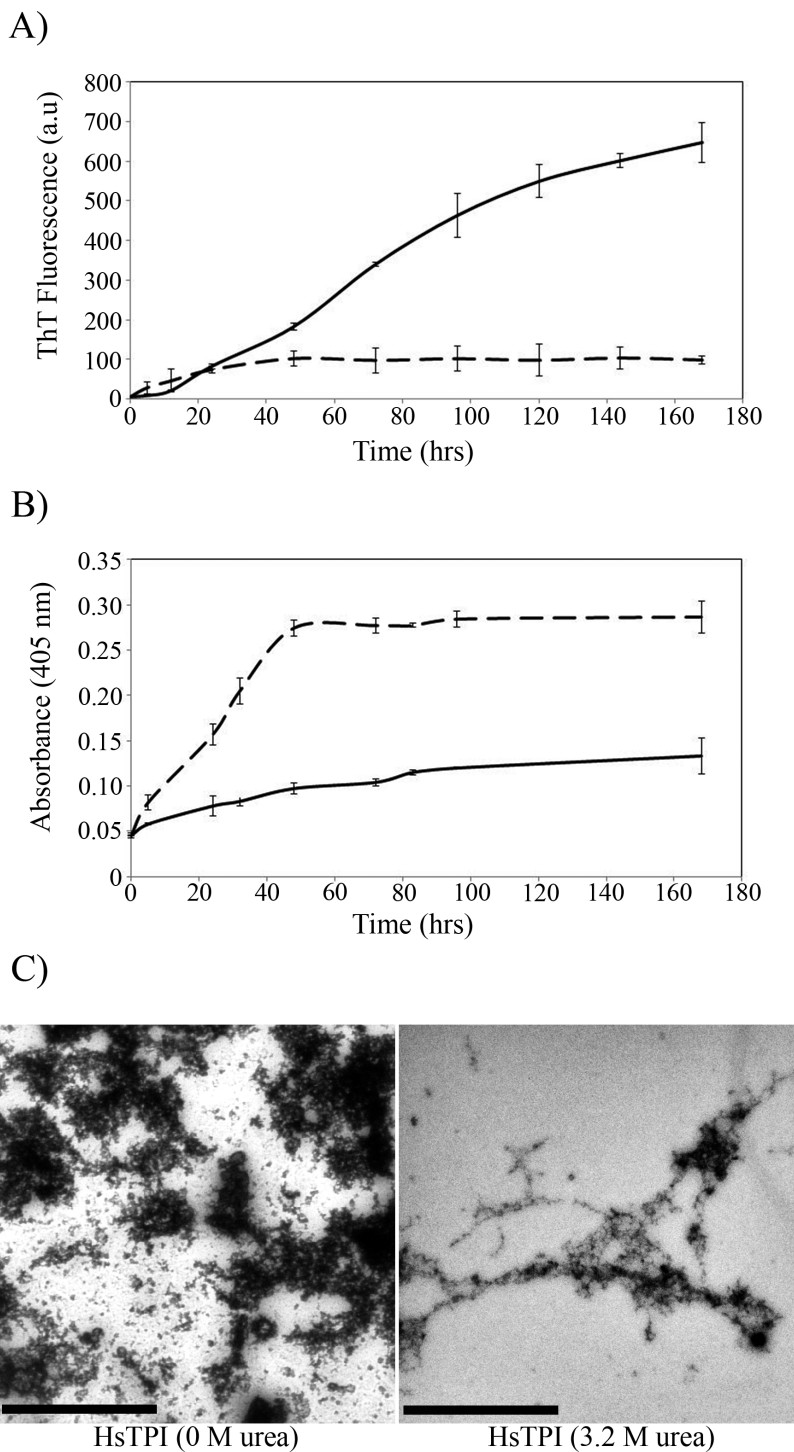

**Figure 1** **Aggregation kinetics followed by (A) ThT fluorescence and (B) by turbidimetry at 405 nm of HsTPI$_n$ (dashed line) and HsTPI$_{urea}$ (solid line). (C) TEM images of HsTPI aggregates at the final time point of aggregation.** Scale bars are 1 M.
A)

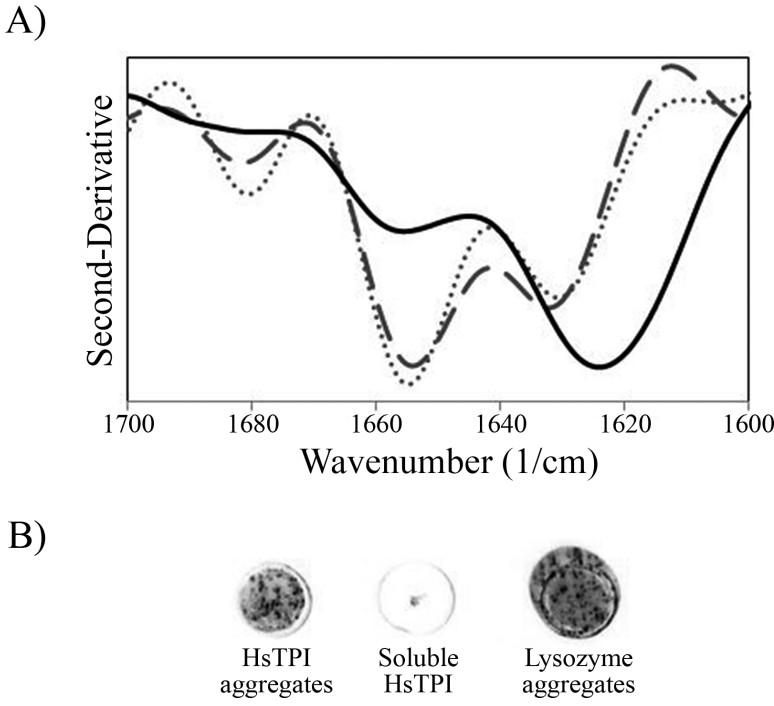

B)

HsTPI
aggregates

Soluble
HsTPI

Lysozyme
aggregates

**Figure 2** **Secondary structure of HsTPI aggregates.** (A) Second-derivative ATR-FTIR spectra in the amide I region of salted-out HsTPI (dotted line), HsTPI$_n$ (dashed line) and HsTPI$_{urea}$ (solid line). (B) Dot-blot assay of HsTPI aggregates with the WO1 antibody confirming cross-$\beta$ structure.

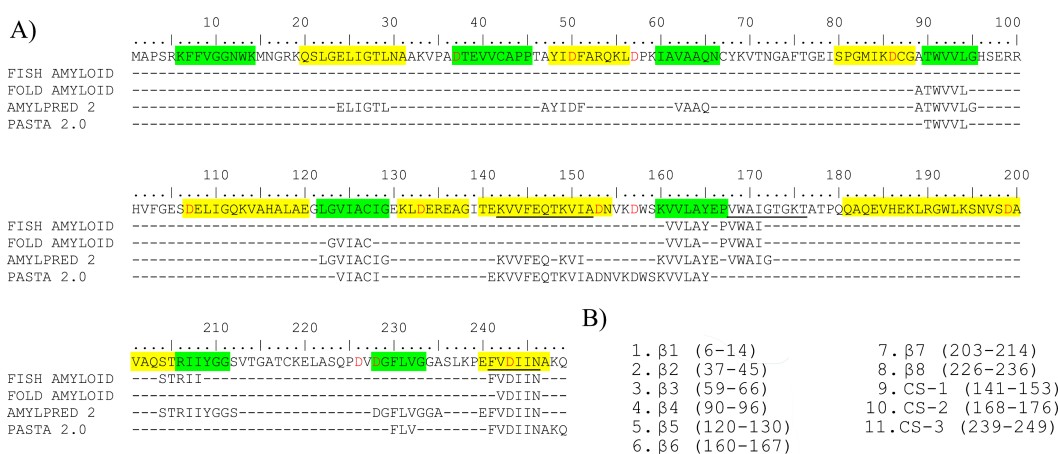

**Figure 3** **Identification of potential fibrillogenic regions in HsTPI.** (A) Consensus of different prediction methods of fibrillogenic regions. The $\beta$-strands are shown in green, the $\alpha$-helices in yellow and the chameleonic sequences are underlined. Aspartic residues are in red, indicating the potential sites for acid hydrolysis. Prediction hits are shown in the corresponding line of the predictor. (B) List of the potential fibrillogenic regions selected for peptide aggregation assay. All $\beta$-strands were selected as well as the three more significant chameleonic sequences of the protein.

## Fibrillogenic regions in HsTPI

In order to identify the fibrillogenic regions in HsTPI, the primary structure was submitted to four servers that use different protein aggregation prediction algorithms: FISH-AMYLOID (*Gasior & Kotulska, 2014*), FOLD-AMYLOID (*Garbuzynskiy, Lobanov & Galzitskaya, 2010*), AMYLPRED 2 (*Tsolis et al., 2013*) and PASTA 2.0 (*Walsh et al., 2014*). The prediction algorithms were based on a database of fibrillogenic sequences of prions, disease-associated proteins and functional amyloid proteins (*Fernandez-Escamilla et al., 2004*), as well as some physical-chemical principles, such as secondary structure propensity, hydrogen-bonding potential, chameleonic sequences (CS), fully buried regions and structure-breaker residues such as proline. The predictions reached a consensus for eight regions primarily located at the C-terminus half of the protein (Fig. 3A). The sequences of these eight regions were selected to carry out aggregation assays (Fig. 3B). These peptides comprise the three chameleonic regions: CS-1 (residues 141–153), CS-2 (residues 168–176) and CS-3 (residues 239–249); and five $\beta$-strand regions $\beta$4 (residues 90–96), $\beta$5 (residues 120–130), $\beta$6 (residues 160–167), $\beta$7 (residues 203–214) and $\beta$8 (residues 226–236). Furthermore, the regions covering the three non-recognized $\beta$-strand regions: $\beta$1 (residues 6–14), $\beta$2 (residues 37–45) and $\beta$3 (residues 59–66), were also included for peptide aggregation assays due their inherent propensity to form parallel $\beta$-sheets and potential $\beta$-aggregation. To maximize the solubility of the synthetic peptides, extra native residues were added to the predicted regions as recommended by GenScript USA Inc (Table S1).

During the peptide aggregation assays, the $\beta$3 peptide became turbid by the second day of incubation followed by the $\beta$6, $\beta$7 and CS-3 peptide, where turbidity appeared by the fifth day. Measurements of the final ThT fluorescence intensities at 485 nm were recorded for all of the samples. The $\beta$3, $\beta$6, $\beta$7 and CS-3 peptide aggregates showed a clear increase in ThT fluorescence indicating $\beta$-aggregation (Fig. 4A). The $\beta$3 aggregates showed the highest ThT fluorescence intensity, indicating a greater fibril formation for this sequence than for the $\beta$6, $\beta$7 and CS-3 sequences. In addition, aggregates of the $\beta$3, $\beta$6, $\beta$7 and CS-3 peptides were tested in the spectrometric birefringence assays with Congo red (*Nilsson, 2004*). The four aggregates exhibited a red-green birefringence, as they displayed a maximal peak at 540 nm in the absorbance spectrum in the range of 400–700 nm, indicative of amyloid formation, (Fig. 4B). The *b* value was consistent with the ThT fluorescence measurements, indicating higher $\beta$-aggregation for $\beta$3 and $\beta$7 peptides. Furthermore, all peptide aggregates were examined by TEM. The four peptide aggregates displayed fibrillar morphology; however, a higher degree of association was achieved by the $\beta$3 peptide since it formed a dense net of mature fibers (Fig. 5). All others studied peptides showed amorphous aggregation or no aggregation at all.

## Acid hydrolysis of HsTPI

It is known that protein processing is an important factor in physiopathology of some amyloid diseases (*O'Brien & Wong, 2011*; *Solomon et al., 2009*). This processing can be limited to specific sites in wild-type proteins, but mutations and environmental factors that destabilize the native-state can increase the fragmentation of a protein, exposing
A)

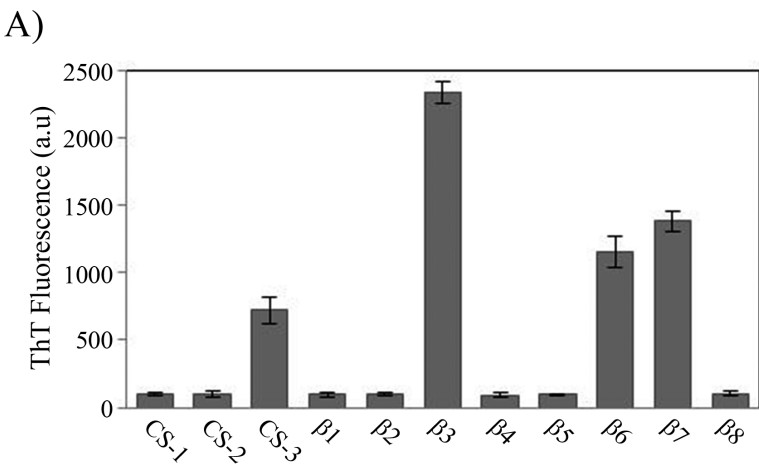

B)

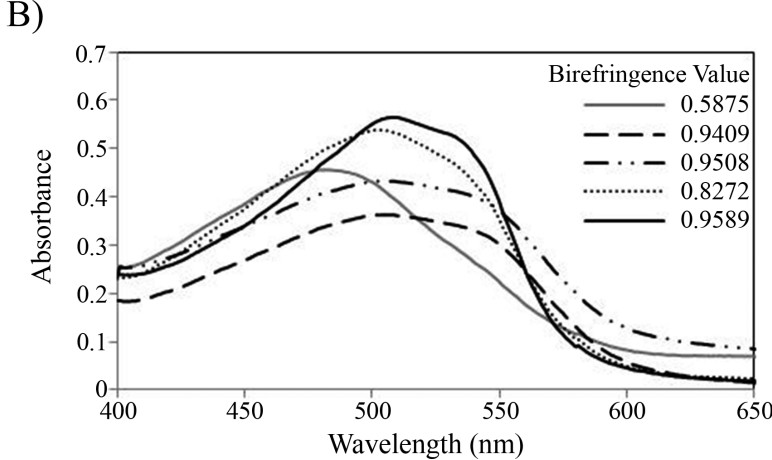

**Figure 4** **Peptide aggregation.** (A) ThT fluorescence intensities at 485 nm of aggregates of peptides at final time point of incubation. (B) Congo red birefringence assay of the β3 (solid black line), β6 (dashed line), β7 (dotted-dashed line) and CS-3 (dotted line) aggregates. A maximal peak at 540 nm is shown in aggregates compared with Congo red alone (solid gray line).

potential amyloid-prone regions. Incubation of proteins under acidic conditions gives rise to random fragmentation near Asp residues (*Li et al., 2001*), and favors amyloid fibril formation (*Frare et al., 2004*; *Mishra et al., 2007*). In order to investigate the potential amyloidogenic regions in HsTPI, a partial acidic hydrolysis was performed by incubating the protein at pH 2 and 65 °C during 8 h (Fig. 6A). Assemblies of high molecular weight appeared upon partial hydrolysis, indicating a rapid association of the resulting fragments. The aggregation of fragmented HsTPI followed a nucleated polymerization mechanism with a lag phase of approximately 45 h that reached saturation after 140 h (Fig. 6B). Fibrillar morphology was confirmed by TEM images (Fig. 6C). The dissociated aggregate, after 7.4 M Gdm-HCl treatment, was analyzed by SDS-PAGE and nLC-MS-MS. Figure 6D shows one main fragment below 10 kDa, a small band above 15 kDa, and other light bands at higher MW, suggesting that more than one fragment were incorporated into the amyloid

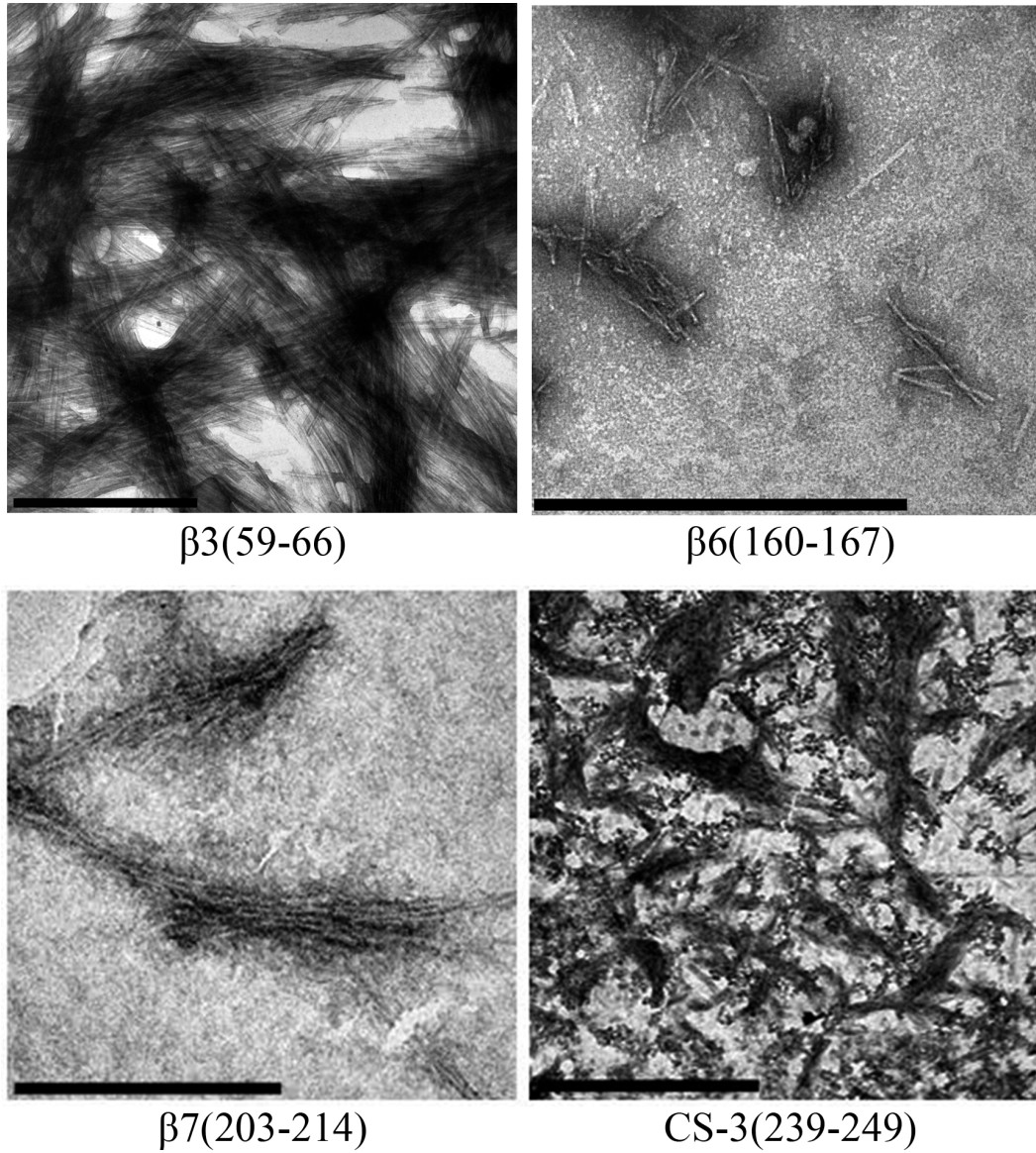

β3(59-66)        β6(160-167)

β7(203-214)        CS-3(239-249)

**Figure 5   TEM images of peptides aggregates.**  The scale bar are 1 μm.

fibril. However, the nLC-MS/MS analysis detected only one fragment with average mass of 3143.52 Daltons (Fig. 6E). We presume that the higher molecular weight bands were oligomeric forms of the fragment not dissociated by Gdm-HCl. The sequence of the band found by nLC-MS-MS corresponds to residues 57–85 from HsTPI, covering the entire $\beta$3-strand and most of the $\alpha$3-helix of the $(\beta/\alpha)_8$ barrel.

As displayed in the resultant sequence, the glutamine 65 was deaminated into a glutamate due to acid treatment of the protein. However, this chemical change had no effect in fibrillogenic propensity since both, the synthetic ($\beta$3 peptide) and hydrolyzed fragment, were able to form amyloid-like fibrils. It is noteworthy that despite the fact that every

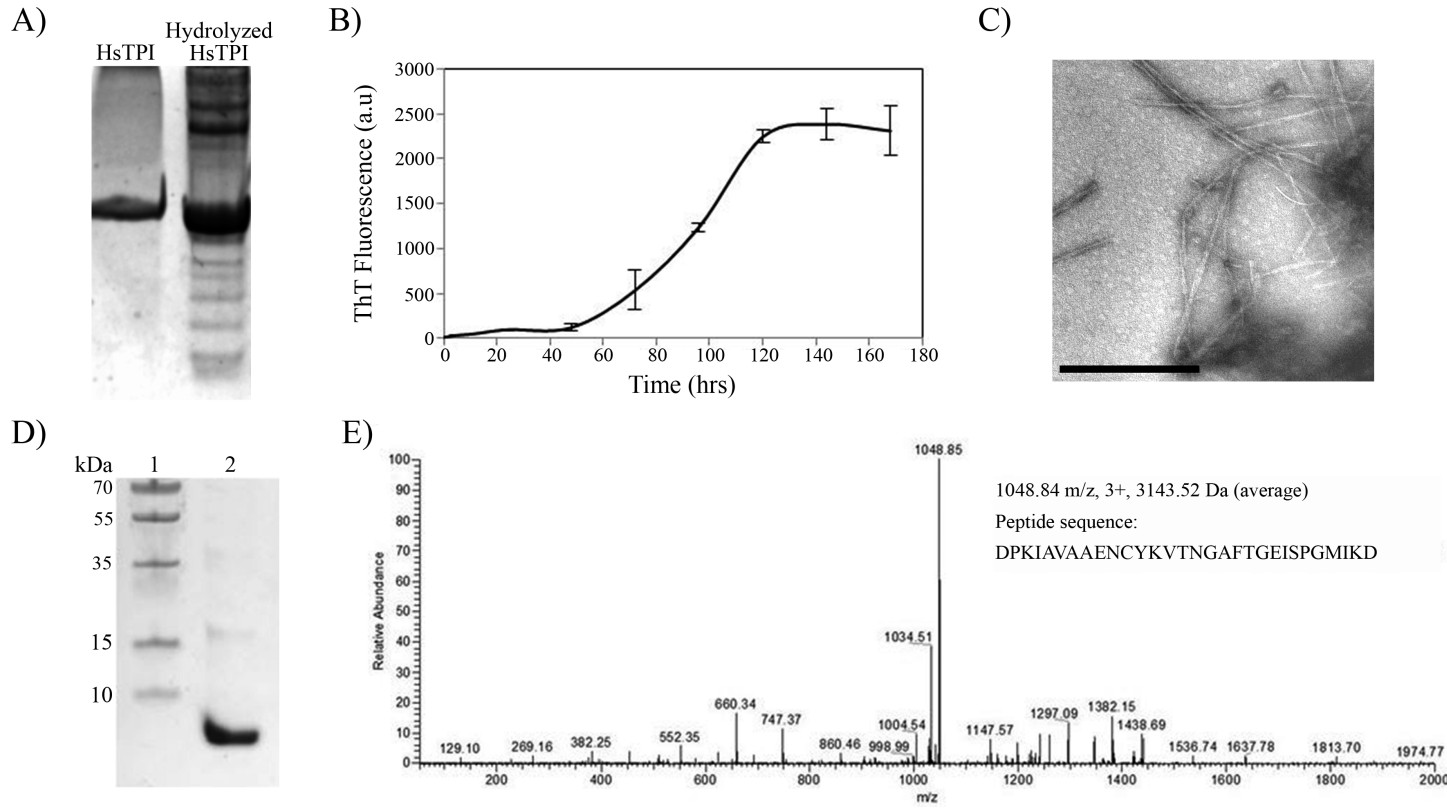

**Figure 6** **Acid hydrolysis of HsTPI.** (A) Tricine SDS-PAGE of hydrolyzed HsTPI. (B) Aggregation kinetics of the hydrolyzed fragments of HsTPI followed by ThT fluorescence. (C) TEM image of the amyloid fibrils formed by fragmented HsTPI; the scale bars is 1 μM. (D) Tricine SDS-PAGE of the enriched fragment upon aggregation. (E) MS/MS spectrum of the triply charged precursor ion at *m/z* 1048.84 identifies the amyloid fragment as the sequence DPKIAVAAENCYKVTNGAFTGEISPGMIKD, which corresponds to residues 57–85 of HsTPI.

aspartic residue was a potential cleavage site (*Li et al., 2001*) and therefore all predicted segments were potentially covered, no other fragment was aggregated upon hydrolysis.

## DISCUSSION

### HsTPI aggregation

The incubation of HsTPI$_n$ led to non-fibrillar aggregates as shown in the TEM images. ATR-FTIR analysis of this aggregates revealed that HsTPI$_n$ aggregates kept most of its native secondary structure. Previous studies have remarked that globular proteins can self-assemble into native-like aggregates promoted by subtle conformational changes not necessarily implying unfolding, that can be on- or off-pathway of fibrillogenesis (*Bemporad & Chiti, 2009*; *Jahn et al., 2006*). The turbidimetric assay showed a fast increase during the first hours of incubation, reaching saturation after 40 h (Fig. 1B). This change is accompanied by a small increase of ThT fluorescence (Fig. 1A). The rapidity with which these aggregates were formed, as well as the low intensity of ThT fluorescence indicate rather an native-like amorphous nature of HsTPI$_n$ aggregates. These results suggest that HsTPI under native-favoring conditions has great stability that prevents $\beta$-aggregation. In

this matter and similarly to other TPIs, HsTPI thermal denaturation follows a two-state irreversible model with a first-order kinetic rate constant of $7.2 \times 10^{-6}$ min$^{-1}$ at 37 °C (*Aguirre et al., 2014*; *Costas et al., 2009*). According to this value, seven days of incubation under native-favoring conditions are insufficient to allow HsTPI to visit conformational states that could lead to fibrillogenesis.

On the other hand, the slightly destabilization of HsTPI structure with 3.2 M urea (*Mainfroid et al., 1996a*) showed an increase in $\beta$-aggregation according to ThT fluorescence intensities (Fig. 1A) suggesting that native state is protected by a high energy barrier that impedes the exploration of intermediate states susceptible to $\beta$-aggregation. Higher concentrations of urea did not increase $\beta$-aggregation (Fig. S2), as expected, since urea can solvate the main-chain and compete for hydrogen bonds during $\beta$-aggregation (*Cai et al., 2014*; *Hamada & Dobson, 2002*; *Zhang et al., 2014*). In addition to ThT fluorescence, HsTPI$_{urea}$ aggregates were analyzed by ATR-FTIR demonstrating formation of new $\beta$-structure with a characteristic lower-frequency band position around 1,624 cm$^{-1}$ indicative of cross-$\beta$ formation (*Moran & Zanni, 2014*; *Zandomeneghi et al., 2004*). Even though recent studies have sighted a clear tendency in new $\beta$-structure formation upon aggregation despite the nature of the aggregate, the position of the band below 1,630 nm is indicative of a stronger H-bond formation as in fibrils (*Shivu et al., 2013*; *Wang et al., 2010*). In this regard, cross-$\beta$ structure was further confirmed by the recognition of WO1 antibody (*O'Nuallain & Wetzel, 2002*).

It is interesting to note that destabilized HsTPI followed a nucleation-based aggregation model with a short lag phase; however, the characteristic exponential fibril elongation phase of the nucleated-polymerization model, was not observed. Instead, the kinetics showed a very slow increase in ThT fluorescence during the seven days of incubation. A similar behavior was observed in the amyloid fibril formation of the SH3 domain of the PI3 kinase, which at pH 3.6 formed amorphous aggregates (1–3 h) with the posterior appearing of curly fibrils (5 days) (*Bader et al., 2006*). It was suggested in this work that amorphous aggregates were energetically more favorable than the nucleation needed for fibril formation. In contrast to HsTPI$_{urea}$, the aggregation of the fibrillogenic fragment found upon acid hydrolysis exhibited a longer lag phase with an exponential elongation of amyloid fibrils in a clear nucleated-polymerization model (Fig. 6B). It seems that the disordered association of HsTPI$_{urea}$ was caused by interactions of the rest of the protein, not present in the fragment (57–85), competing with cross-$\beta$ association, delaying fibrillogenesis. Nevertheless, these interactions accelerated the intermolecular association in HsTPI$_{urea}$ exhibiting a shorter lag phase. It has been described as a similar cooperativity in both amorphous and $\beta$-sheet oligomerization, suggesting that disorder aggregation could compete with $\beta$-aggregation in early steps of fibril formation (*Hills Jr & Brooks, 2007*; *Krishnan & Raibekas, 2009*; *Vetri et al., 2007*). Moreover, this early amorphous aggregates can play an important role in the recruitment and association of protein molecules into cross-$\beta$ structure by conformational conversion (*Auer et al., 2008*; *Johnson et al., 2012*; *Serio et al., 2000*). This proposed mechanism can explain the co-aggregation of disordered structures with poor fibrillar morphology shown by HsTPI$_{urea}$ aggregates, indicating that the fibrillogenesis was on track but it was incomplete after 7 days of incubation.

## Fibrillogenic region of HsTPI

The identification of fibrillogenic regions was achieved through different algorithms. From these, eight consensus predictions were found, from which only 3 peptides covering the regions $\beta6$, $\beta7$ and CS-3, respectively, were able to form amyloid-like aggregates as confirmed by fluorescence and birefringence assays (Fig. 4). In particular, the $\beta$-aggregation of the $\beta7$ peptide was consistent with previous evidence of amyloid-like aggregation in the region containing the equivalent strand in the *Escherichia coli* TPI (*Contreras et al., 1999*). However, *Contreras et al. (1999)* did not delimit the cross-$\beta$ core of the 32 residue fragment. According to our results, we can infer that $\beta7$-strand is at least one cross-$\beta$ core in the *E. coli* TPI fibrillogenic fragment (186–218) due to its high identity with $\beta7$-strand from HsTPI.

In addition to the consensual predicted regions, we found that the sequence comprising the $\beta3$-strand (residues 59–66), only predicted by AMYLPRED 2 server (*Tsolis et al., 2013*), was highly fibrillogenic. All others studied peptides showed amorphous aggregation or no aggregation at all. It is interesting that from an independent experiment in which fibrillogenesis was investigated from acid-generated fragments of the protein, the only fragment detected from dissociated fibers contains precisely the $\beta$-3 strand sequence, corroborating the great fibrillogenic propensity of this region.

## Amyloid protective features in HsTPI

The presence of at least 4 cross-$\beta$ regions, including one with high fibrillogenic propensity, in a highly expressed protein that participates in the central metabolism of any living cell, raises the question of how nature has avoided the major catastrophic events that could preclude the necessary balance to sustain life. Some structural characteristics of HsTPI topology could be consider as protective features.

First, all parallel $\beta$-stands are buried inside the protein, forming a $\beta$-barrel that prevents further $\beta$-sheet propagation and, thereby, $\beta$-aggregation. In order to form cross-$\beta$ structure, the fibrillogenic $\beta$-strands found in this study ($\beta3$, $\beta6$ and $\beta7$) must be exposed to the surface. However, because the amino acids constituting the $\beta$-barrel are predominantly hydrophobic, the exposure of the inner core may collapse into amorphous aggregates instead of rapidly forming cross-$\beta$ structures, retarding or avoiding amyloid-like aggregation.

In the case of solvent-exposed regions, which are more accessible for intermolecular associations, there is a structural restriction avoiding the $\beta$-conformation that promotes the $\alpha$-helix or random coil conformations. This mechanism could prevent the cross-$\beta$ association of CS-3 in the native fold because this region is restricted to the last $\alpha$-helix of HsTPI.

It is noteworthy that most of the predicted and demonstrated fibrillogenic regions are in the C-terminus half of the protein. Recent studies of *in vivo* folding indicate that folding starts as soon as the polypeptide chain leaves the ribosome tunnel (*O'Brien et al., 2010*). So that *in vivo*, by the time the C-regions are exposed, the N-terminus half has probably started its folding process, directing the folding of the remaining of the polypeptide, and avoiding any off-pathway intermolecular interaction. Once the native state is reached

its high energy barrier protects it from partial unfolding that could expose these regions (*O'Brien et al., 2011*; *Ugrinov & Clark, 2010*).

In addition to the tertiary structure of $(\beta/\alpha)_8$, TPI is always found as an oligomer, and more frequently as a dimer. This intermolecular association contributes to an 8-fold increase in the stability of the human enzyme (*Mainfroid et al., 1996a*). The interface region is precisely formed by the loop following the $\beta$3-strand that interdigitates into the active site of the other subunit. This interaction gives extra protection to the region around the $\beta$3-strand. Although most of the efforts to perturb the dimeric interface of the protein have yielded inactive proteins (*Borchert et al., 1994*; *Borchert et al., 1995*; *Schliebs et al., 1997*), the generation of a sufficiently active monomeric variant (*Saab-Rincon et al., 2001*) rules out the possibility that activity is the only major selective pressure for this protein to maintain its oligomeric state. Instead, it is possible that the changes in tertiary contacts in the fibrillogenic regions upon dimerization, increases the energetic barrier for the formation of amyloid fibrils, as suggested by *Buell et al. (2012)*, which could be another selective factor to maintain TPIs as dimers.

Finally, it has been observed that the size of polypeptide chain could influence in the aggregation of proteins (*Baldwin et al., 2011*; *Ramshini et al., 2011*; *Solomon et al., 2009*). The length of the polypeptide chain could be associated with the number of possible conformation states in the intermolecular protein association increasing the number and complexity of the aggregation pathways. Since HsTPI is a medium-size protein with a compact, stable and evolution-selected topology is reasonably to speculate the existence of competitive aggregation pathways, once the native state is altered, that avoid cross-$\beta$ formation.

## CONCLUSION

It is clear that albeit containing at least four potential amyloidogenic regions, the nature of HsTPI confers protection against the formation of toxic amyloid aggregates. However, mutations or post-translational modifications might affect its solubility, stability and/or folding, allowing it to develop a role in amyloid diseases.

## ACKNOWLEDGEMENTS

We thank Dr. Ronald Wetzel and Ravindra Kodali for kindly donating the WO1 antibody, Dr. Armando Gómez Poyou† for providing the pET3a-HsTPI plasmid, Biol. Filiberto Sanchez López for technical support, and the following facilities of the Instituto de Biotecnología, Universidad Nacional Autónoma de México for their services at different stages of this work: Laboratorio Nacional de Microscopía Avanzada, specifically, Dr. Guadalupe Zavala; Laboratorio Universitario de Proteómica, specifically Dr. Cesar Ferreira Batista; and Unidad de Cómputo and Unidad de Biblioteca.

### Funding

This work was supported by the Programa de Apoyo a Proyectos de Investigación e Innovación Tecnológica (PAPIIT) (grant number IN211414 to GSR) and the Consejo Nacional de Ciencia y Tecnología (CONACYT) (grant number 154194). The funders had no role in study design, data collection and analysis, decision to publish, or preparation of the manuscript.

### Grant Disclosures

The following grant information was disclosed by the authors:
Programa de Apoyo a Proyectos de Investigación e Innovación Tecnológica (PAPIIT): IN211414.
Consejo Nacional de Ciencia y Tecnología (CONACYT): 154194.

### Competing Interests

The authors declare there are no competing interests.

### Author Contributions

- Edson N. Carcamo-Noriega performed the experiments, analyzed the data, wrote the paper, prepared figures and/or tables.
- Gloria Saab-Rincon conceived and designed the experiments, contributed reagents/materials/analysis tools, wrote the paper, reviewed drafts of the paper.

### Data Availability

Data can be found in Data S1

### Supplemental Information

Supplemental information for this article can be found online at http://dx.doi.org/10.7717/peerj.1676#supplemental-information.

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
