# Peer review of "Identification of fibrillogenic regions in human triosephosphate isomerase"

_PeerJ, doi:10.7717/peerj.1676_

## Round 0.1 · original submission · Major Revisions

Please address critical issues raised by the reviewers and revise the manuscript accordingly.

Reviewer 1 ·

Basic reporting

Basic structure of the article is fine. There are a few grammatical errors.

Experimental design

Overall the experimental design is ok but there are some problems. TEM images are of poor quality due to overstaining and fairly low magnification. It is especially important in Fig. 1 b/c it is difficult to tell the shape of the aggregates there. There is also a problem with Fig. 2B as dots in the dot blot look strange. Also, there isn’t enough data on aggregation of the protein in the absence of urea. While FTIR of the aggregates is useful, perhaps kinetic analysis (e.g. by static or dynamic light scattering) could show what the kinetics of the process is. I also think the authors need to do a better job explaining the reason for using the proteolytic digest of the protein for aggregation.

Validity of the findings

Other than poor quality TEM images, the finding appear to be valid.

Reviewer 2 ·

Basic reporting

This article “Identification of fibrillogenic regions in human triosephosphate isomerase (TPI)” by Carcamo-Noriega and Saab-Rincon aims to locate the fibrillation inducing regions in TPI.

I have two main points that I want to mention
1) The use of HEWL in figure1 masks the real data of TPI aggregation, there appears to be an elongation and plateau phase in TPI curve although of lower intensity. There is no reason ThT fluorescence of TPI aggregation should match with HEWL fluorescence neither is there any comparison. Re-plotting the TPI data will therefore change the whole context and reference.
2) In figure6 authors show hydrolyzed product before aggregation (6A lane 2) and after aggregation (6D lane2) and conclude the ~3kDa fragment in 6D constitutes fibrils. But we don’t see this fragment in the starting material (6A), there are only higher fragments. On the contrary, we see shades of higher fragments in 6D. Further, since they started from a mixture of fragments how do authors conclude the 3kDa is not a mixture or various fragment rather than strand b3. Fibrillation of as small as hexa-peptides has been reported widely in literature.

3) minor comment - language, typographical mistakes.

I would suggest a major revision.

Experimental design

Please see basic reporting section

Validity of the findings

Please see basic reporting section

Additional comments

Please see basic reporting section

---

## Round 0.2 · accepted · Accept

Thank you for the attentive consideration of the reviewers' comments and for the careful revision of the manuscript.